# Rapid Identification of a Genomic Region Conferring Dwarfism in Rapeseed (*Brassica napus* L.) YA2016-12

**Liang Chai** [1],[†] , **Haojie Li** [1],[†], **Jinfang Zhang** [1], **Lintao Wu** [2], **Benchuan Zheng** [1], **Cheng Cui** [1], **Jun Jiang** [1], **Shangqi Zuo** [3] **and Liangcai Jiang** [1],*

1   Crop Research Institute, Sichuan Academy of Agricultural Sciences, Chengdu 610066, China; chailiang1982@126.com (L.C.); lhjie16@163.com (H.L.); zhangjinfang567@163.com (J.Z.); zhengbenchuan1987@163.com (B.Z.); cuicheng005@163.com (C.C.); jiangjun227@163.com (J.J.)
2   Rape Research Institute, Guizhou Academy of Agricultural Sciences, Guiyang 550008, China; wult12@126.com
3   Sichuan Kele Rape Research and Development Co., Ltd., Deyang 618000, China; kele780@vip.sina.com
*   Correspondence: jlcrape@163.com; Tel.: +86-28-84504235
†   These authors contributed equally to this work.

**Abstract:** Plant height is a vital agronomic trait for crops, including oilseed crops such as rapeseed (*Brassica napus* L.). It affects the crop yield, oil content, and lodging resistance in rapeseed. In this study, we investigated a dwarf trait controlled by a semi-dominant allele in rapeseed. A dwarf line, YA2016-12, was crossed with a tall line, G184-189, and an $F_2$ population was established. Forty of the tallest plants and 40 of the shortest plants from the $F_2$ population were selected and two DNA pools (tall and dwarf) were constructed by the bulked segregant analysis (BSA) method. The two DNA pools and two parental DNAs were then re-sequenced. A sliding window analysis was used to calculate the Δ(SNP-index) and discover an association region on chromosome A03 with a length of 12.4 Mb. Within this region, we found 1225 genes, including 811 genes with non-synonymous or frameshift mutations between YA2016-12 and G184-189. Alignment to known plant height-related orthologs in *Arabidopsis thaliana*, as well as KEGG pathway and gene ontology annotations, was used to identify nine candidate genes (*BnaA03g31770D, BnaA03g37960D, BnaA03g24740D, BnaA03g40550D, BnaA03g26120D, BnaA03g35130D, BnaA03g42350D, BnaA03g25610D,* and *BnaA03g39850D*) involved in gibberellin or cytokinin signaling. Identification of the causal gene for this trait, and of genetic markers linked to favorable alleles, has potential utility for marker-assisted selection to breed rapeseed varieties with improved height.

**Keywords:** association analysis; dwarf; plant height; rapeseed (*Brassica napus* L.)

## 1. Introduction

Rapeseed (*Brassica naups* L.) is a major oil crop globally. It is an allotetraploid (AACC, $2n = 38$) with a complex genome. Heterosis is used widely in rapeseed breeding, which has resulted in plants that are stronger and taller [1] at the vegetative stage. Plant height is an important yield-related trait; taller plants are more prone to lodging, which results in decreased yield due to grain loss and spoilage, and also makes mechanized harvesting difficult [2]. In some crops, such as wheat (*Triticum* sp.) and rice (*Oryza sativa*), dwarf or semi-dwarf plant types were introduced in the 1960s and 1970s, resulting in increased yields and the so-called "green revolution" [2]. Therefore, breeding rapeseed for dwarfism or semi-dwarfism plant stature could increase lodging resistance, stabilizing production and facilitating mechanized harvesting.

Plant height in crops is generally considered to be a quantitative trait controlled by multiple loci and significantly affected by environmental factors [3]. Li et al. [4] used 472 rapeseed accessions to

conduct a genome-wide association study (GWAS) and identified eight quantitative trait loci (QTLs) for plant height on chromosomes A03, A05, A07, and C07. Mei et al. [5] used a segregating population of rapeseed with 145 $F_{2:3}$ lines, which was obtained by crossing a tall/later flowering line with a dwarf/early flowering line, and found seven QTLs related to plant height. These QTLs accounted for 8.5–28.6% of the phenotypic variation, and two of them were identified in both years. Shi et al. [6] used a doubled-haploid population with 188 lines and found 21 QTLs related to plant height under phosphorus deficiency conditions, which decreased plant height.

In other instances where the dwarfism trait was obtained from mutation, dwarfism was found to be controlled by one major gene. Wang et al. [7] fine-mapped an ethyl methane sulfonate (EMS) mutagenized line with a dominant locus controlling dwarfism to linkage group A09 using single-nucleotide polymorphism (SNP) markers. Liu et al. [8] investigated a semi-dwarf mutant phenotype and found that it was caused by a missense mutation in a DELLA protein. Wang et al. [9] and Li et al. [10] studied a dwarf rapeseed mutant 'NDF-1', in which dwarfism is controlled by a major gene with a mainly additive and non-significant dominance effect, and revealed that three bases mutated in the pyrimidine box (P-box) of the *Gibberellin Insensitive Dwarf 1* (*BnGID1*) promoter changed its expression. To date, many height-related loci/genes involved in hormone biosynthesis or signal transduction in crops have been identified, especially genes involved in gibberellin (GA) signal transduction: *Reduced Height* (*Rht*) [11] in wheat, orthologs of *Arabidopsis thaliana GIBBERELLIN INSENSITIVE* (*GAI*); *Semidwarfing Gene* (*sd1*) in rice, which encodes a mutant enzyme involved in gibberellin synthesis [12–14]; and *ZmGA3ox2* in maize (*Zea mays*) [15], an ortholog of *OsGA3ox2*, which encodes a GA3 β-hydroxylase.

Bulked segregant analysis (BSA) [16] was traditionally combined with markers such as sequence-related amplified polymorphism (SRAP) [17], simple sequence repeats (SSR) [18], and random amplified polymorphic DNA (RAPD) [19] to locate the trait-related regions on chromosomes. More recently, next-generation sequencing technologies, such as whole-genome re-sequencing (WGR), have been commonly used to identify key loci related to agronomic traits. The sequencing of the *B. napus* genome [20] facilitated great advances in the identification of key genes related to branch angle [21], seed weight [22], and petal color [23], among others, in rapeseed.

In this study, a dwarfism trait in the dwarf rapeseed line YA2016-12, which was first discovered in *B. napus* × *B. rapa* hybrid offspring, was crossed with the tall rapeseed line G184-189. Using plants from the $F_2$ population, we constructed two DNA pools (tall pool and dwarf pool) by the BSA method and performed WGR. A sliding window analysis was used to identify the trait-associated regions from the SNP and insertion/deletion (InDel) data. Genes within the associated region were then aligned with known plant height-related genes in *Arabidopsis*, and their annotations were also analyzed. Finally, we selected nine potential candidate genes for this dwarfism trait in rapeseed.

## 2. Materials and Methods

### 2.1. Plant Materials

The rapeseed dwarfism phenotype studied here was originally discovered in *B. napus* × *B. rapa* hybrid offspring. The dwarf plant was then self-pollinated for several generations until the homozygous, stable line YA2016-12 was obtained. Both the tall line G184-189 and the dwarf line YA2016-12 were kept in the Crop Science Institute, Sichuan Academy of Agricultural Sciences (SAAS). YA2016-12 plants were crossed with G184-189 plants to generate the $F_1$ population, followed by self-pollination of the $F_1$ plants to obtain the $F_2$ population. The parental plants and the $F_1$ and $F_2$ populations were grown in the experimental field of SAAS under natural conditions in Xindu District, Chengdu, Sichuan Province.

## 2.2. Phenotypic Data Collection and Genetic Analysis

Individual plants were measured at the final-flowering stage. Plant height (PH), first primary branch height (FPBH), and main inflorescence length (MIL) were measured for each plant and the branch segment length (BSL) was then calculated. PH was defined as the distance from the soil surface to the top of the main inflorescence; FPBH was defined as the distance from the soil surface to the first primary branch with siliques; MIL was defined as the distance from the top of the main inflorescence to its closest primary branch with siliques; BSL = PH − FPBH − MIL [24]. The experiments were replicated three times.

Measurements were recorded for 10 plants of each parental plant (G184-189 and YA2016-12), with the $F_1$ population consisting of 20 individuals and the $F_2$ population consisting of 194 individuals. Then a Chi square test was carried out in MS Excel 2013.

## 2.3. Whole-Genome Re-Sequencing Library Construction and High-Throughput Sequencing

After the measurements, the 40 tallest plants and the 40 shortest plants in the $F_2$ segregating population, as well as the parental plants G184-189 and YA2016-12, were sampled and their DNA was isolated from leaves by the CTAB (Cetyl Trimethyl Ammonium Bromide) method [25]. The DNA was diluted to 50 ng/µl with an OD260/280 value ranging from 1.8 to 2.2. RNase A was used to remove RNA contamination. Equal amounts of DNA from the plants in each group were mixed to form two pools: the tall-pool (T-pool) and the dwarf-pool (D-pool). DNA of the two pools and the two parental plants were then utilized for pair-end sequencing library construction. The sequencing was conducted on the Illumina HiSeq 4000 platform (Illumina, Inc.; San Diego, CA, USA).

## 2.4. Filtering of Clean Reads and Alignment to the Reference Genome

Reads obtained from sequencing included raw reads containing adapters or low-quality bases which would affect the following assembly and analysis. Thus, to obtain only high-quality clean reads, reads were further filtered according to the following rules: (1) remove reads containing adapters; (2) remove reads containing more than 10% of unknown nucleotides (N); (3) remove reads containing more than 50% of low quality ($Q$-value $\leq$ 20) bases.

The software BWA [26] was used to align the clean reads from each sample to the *B. napus* reference genome [20], with the settings 'mem 4-k 32-M', where -k is the minimum seed length, and -M is an option used to mark shorter split alignment hits as secondary alignments. Alignment files were converted to SAM/BAM files using SAMtools [27].

## 2.5. Identification and Annotation of Variants

Variant calling was performed for the multi-sample using the Genome Analysis Toolkit (GATK) [28] Unified Genotyper. SNPs and InDels were filtered using GATK's Variant Filtration with the following standards: -Window 4, -filter "QD < 4.0 ‖ FS > 60.0 ‖ MQ < 40.0 ", -G_filter "GQ < 20" and those exhibiting segregation distortion or sequencing errors were discarded. In order to determine the physical positions of each variant, the software tool ANNOVAR [29] was used to align and annotate SNPs or InDels.

## 2.6. Association Analysis

A sliding window analysis was applied to the frequency distribution of SNPs (SNP-index) in the population of bulked individuals, and the SNP-index was calculated for all the SNP positions. SNPs differing between YA2016-12 and G184-189 but homozygous within each parent with a read depth >2 were kept; while SNP positions with an SNP-index <0.3 or >0.7, or a read depth <7 from the two sequences were excluded, as these may represent spurious SNPs called due to sequencing and/or alignment errors. The Δ(SNP-index) was calculated on the basis of subtraction of the SNP-index between the two bulked pools. QTLs were identified in these positive or negative peak regions with a

95% confidence interval in 10,000 bootstrap replicates. Then, selected SNPs and InDels in the peak regions were annotated to screen for potential functional variants.

## 2.7. Identification of Candidate Genes

Candidate genes were identified by two methods: (1) orthologs in *Arabidopsis*—genes were aligned with the known 143 plant height-related genes in *Arabidopsis* gathered by Shi et al. [30] and detailed information was analyzed; (2) annotations—all genes identified in the association region were searched according to the National Center for Biotechnology Information (NCBI) non-redundant protein database (Nr), Swiss-Prot, TrEMBL, and Cluster of Orthologous Groups of proteins (COG) using BLASTX, and against the NCBI nucleotide database (Nt) using BLASTN, with a cut-off E-value of $10^{-5}$ for both. The gene ontology (GO) annotations and the distribution of gene functions for each gene were obtained by the Blast2GO program [31] and WEGO [32] software. Assignment of genes to different pathways was conducted by using the Kyoto encyclopedia of genes and genomes (KEGG databases) with BLASTX and the KEGG automatic annotation server [33].

## 3. Results

### 3.1. Plant Height Identification and Genetic Analysis

To investigate the inheritance of the dwarf trait in the YA2016-12 line, we crossed YA2016-12 with the homozygous tall line G184-189 to obtain $F_1$ seeds. $F_1$ plants were self-pollinated to generate $F_2$ seeds. The tall parent G184-189 was 154.6 cm on average and the dwarf parent YA2016-12 was 79.6 cm on average (Table 1). The $F_1$ population consisted of 20 individuals and all showed a semi-dwarf phenotype (95 cm on average, Figure 1, Table 1). The 194 individuals in the $F_2$ population could be classified into three different morphologies according to their height: 48 were tall, 100 were semi-dwarf, and the remaining 46 individuals were dwarf (Figure 2, Table 1), which fit the expected ratio of 1:2:1 for a semi-dominant allele, with $X^2 = 0.2$, much smaller than $X_{0.05}^2 = 5.99$. Therefore, the Chi square test indicated that the dwarfism trait is controlled by a single semi-dominant allele. Moreover, we analyzed the correlation of PH to FPBH, MIL, and BSL based on the $F_2$ population and found that the correlation coefficient between PH and BSL was 0.912346 (Table 2), implying a strong positive correlation between PH and BSL; no significant correlation was found between PH and FPBH or MIL.

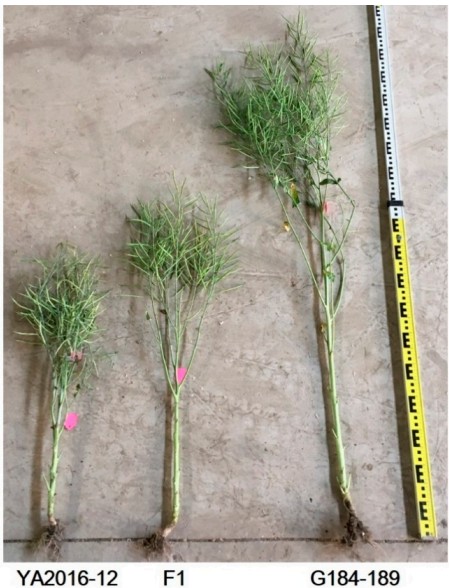

**Figure 1.** Plant height of dwarf parental YA2016-12, tall parental G184-189, and an $F_1$ specimen. The $F_1$ plant represents the semi-dwarf phenotype.

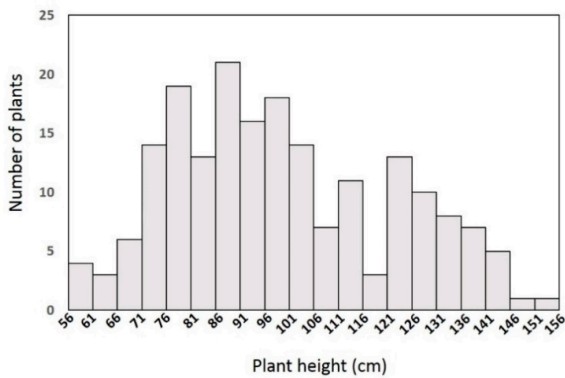

**Figure 2.** Plant height in YA2016-12 × G184-189 F$_2$ population.

**Table 1.** Plant height (cm) of dwarf parental line YA2016-12, tall parental line G184-189, and individuals in the F$_2$ population (mean ± SD). Three replicates were conducted.

| YA2016-12 | G184-189 | F$_2$ Population | | |
| --- | --- | --- | --- | --- |
| | | Dwarf | Semi-Tall | Tall |
| 79.6 ± 5.2 | 154.6 ± 10.8 | 72.4 ± 6.2 | 96.2 ± 9.6 | 130.9 ± 8.4 |

**Table 2.** Correlation of plant height to first primary branch height, main inflorescence length, and branch segment length based on the F$_2$ population.

| | PH | FPBH | MIL | BSL |
| --- | --- | --- | --- | --- |
| PH | 1 | | | |
| FPBH | 0.167175 | 1 | | |
| MIL | 0.327721 | −0.28882 | 1 | |
| BSL | 0.912346 | −0.12794 | 0.13956 | 1 |

PH: Plant height; FPBH: first primary branch height; MIL: main inflorescence length; BSL: branch segment length. The correlation coefficient between PH and BSL was 0.912346, indicating that PH was significantly positive-correlated to BSL. No correlation was found between PH and FPBH or MIL.

### 3.2. Whole-Genome Re-Sequencing Data and Polymorphic Analysis

To map the causal allele, we conducted WGR on four samples—YA2016-12 (dwarf), G184-189 (tall), the extremely tall bulked DNA pool (T-pool), and the extremely dwarf bulked DNA pool (D-pool). High-throughput WGR resulted in a total of 225 Gb of raw data, including over 1.5 billion reads with a read length of 150 bp. After filtering out poor-quality sequence and adaptors, each sample had an average of approximately 339,000,000 high-quality (HQ) clean reads. The Q30 ratio was 93.14% and the GC content was 48.00% on average (Table 3). When aligned to the reference genome, the average sequencing depth was 37.5× and the average number of mapped reads of the four samples were approximately 228 M (Table 3). The four samples produced 9,702,407 SNPs and 1,812,949 InDels (Table 4).

When we aligned our sequences with the reference genome, we found 619,680, 591,036, 825,591, and 827,505 genic (including exonic and intronic) SNPs in the YA2016-12, G184-189, D-pool, and T-pool samples (Figure 3a, Supplementary Table S1), respectively, which resulted in 140,767, 134,453, 188,031, and 188,391 changes in amino acid sequences, respectively (including non-synonymous single nucleotide variations, stopgain, and stoploss) (Figure 4a, Supplementary Table S2). We also found 100,582, 97,925, 132,753, and 133,069 genic InDels in the YA2016-12, G184-189, D-pool, and T-pool samples (Figure 3b, Supplementary Table S3), respectively, which would alter the length of the encoded proteins, causing frameshifts, non-frameshifts, stopgain, and stoploss (Figure 4b, Supplementary Table S4). In general, YA2016-12 showed more polymorphisms than G184-189.

**Table 3.** Summary of the sequencing data for each sample.

| Sample | Number of Clean Reads | Number of HQ Clean Reads | HQ Clean Data (bp) | Number of Mapped Reads (%) | Q30 (%) | GC (%) | Depth (×) |
|---|---|---|---|---|---|---|---|
| D-pool | 375,535,802 | 353,895,666 | 52,680,436,064 | 237,973,082 (67.24%) | 93.39 | 47.88 | 39.14 |
| Dwarf | 381,503,244 | 345,471,474 | 51,521,215,975 | 196,574,422 (56.90%) | 93.31 | 48.04 | 32.33 |
| T-pool | 357,594,574 | 343,756,000 | 51,189,879,695 | 281,686,149 (81.94%) | 92.89 | 46.16 | 46.32 |
| Tall | 386,991,078 | 314,086,614 | 46,850,252,284 | 195,696,255 (62.31%) | 92.98 | 49.91 | 32.19 |

Tall: tall parental G184-189; Dwarf: dwarf parental YA2016-12; T-pool: bulked DNA pool for extremely tall individuals from the $F_2$ population; D-pool: bulked DNA pool for extremely dwarf individuals from the $F_2$ population. HQ: high-quality, as mentioned above; GC: Guanine and cytosine.

**Table 4.** Numbers of single-nucleotide polymorphisms (SNPs) and insertions/deletions (InDels) for each sample.

| Sample | Number of SNPs | Number of InDels |
|---|---|---|
| D-pool | 2,770,948 | 513,479 |
| T-pool | 2,782,281 | 513,926 |
| Dwarf | 2,054,075 | 393,360 |
| Tall | 2,095,103 | 392,184 |

Tall: tall parental G184-189; Dwarf: dwarf parental YA2016-12; T-pool: bulked DNA pool for extremely tall individuals from the $F_2$ population; D-pool: bulked DNA pool for extremely dwarf individuals from the $F_2$ population.

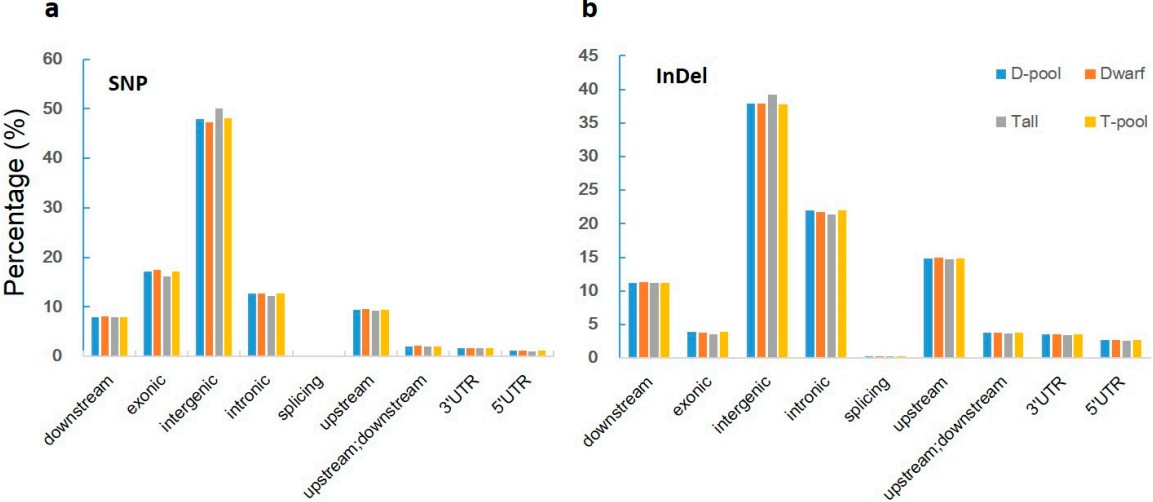

**Figure 3.** Percentage of SNPs and InDels in various locations for each sample. Tall: tall parental line G184-189; Dwarf: dwarf parental line YA2016-12; T-pool: bulked DNA pool for extremely tall individuals from the $F_2$ population; D-pool: bulked DNA pool for extremely dwarf individuals from the $F_2$ population. Downstream: variant overlaps 1-kb region downstream of transcription end site; exonic: variant overlaps a coding exon; intergenic: variant is in an intergenic region; intronic: variant overlaps an intron; splicing: variant is within 2 bp of a splicing junction; upstream: variant overlaps 1-kb region upstream of transcription start site; 3′ UTR: variant overlaps a 3′ untranslated region; 5′ UTR: variant overlaps a 5′ untranslated region. (**a**): Percentage of SNP in various locations; (**b**): Percentage of InDel in various locations.

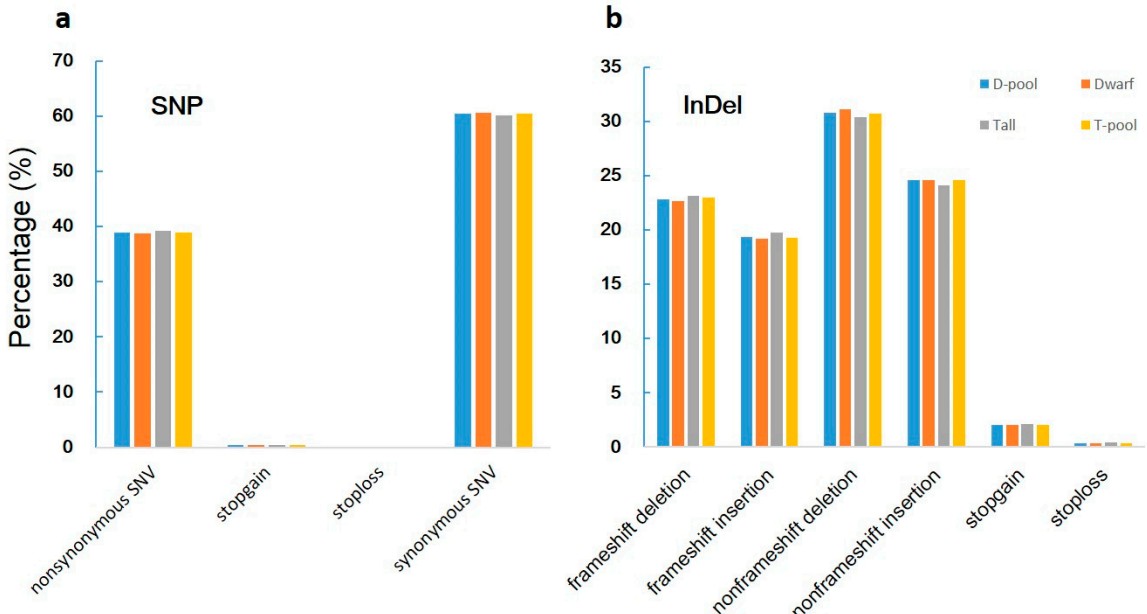

**Figure 4.** Percentage of SNPs and InDels with different function in each sample. Tall: tall parental G184-189; Dwarf: dwarf parental YA2016-12; T-pool: bulked DNA pool for extremely tall individuals from F$_2$ population; D-pool: bulked DNA pool for extremely dwarf individuals from F$_2$ population. Nonsynonymous single nucleotide variants (SNV): a single nucleotide change that cause an amino acid change; stopgain: a nonsynonymous SNV, frameshift insertion/deletion, nonframeshift insertion/deletion, or block substitution that leads to the immediate creation of a stop codon at the variant site. For frameshift mutations, the creation of a stop codon downstream of the variant is not counted as "stopgain"; stoploss: a nonsynonymous SNV, frameshift insertion/deletion, nonframeshift insertion/deletion, or block substitution that leads to the immediate elimination of a stop codon at the variant site; synonymous SNV: a single nucleotide change that does not cause an amino acid change; frameshift deletion: a deletion of one or more nucleotides that causes frameshift changes in a protein coding sequence; frameshift insertion: an insertion of one or more nucleotides that causes frameshift changes in a protein coding sequence; nonframeshift deletion: a deletion of three or multiples of three nucleotides that does not cause frameshift changes in a protein coding sequence; nonframeshift insertion: an insertion of three or multiples of three nucleotides that does not cause frameshift changes in a protein coding sequence;. (**a**): Percentage of SNP with different function in each sample; (**b**): Percentage of InDel with different function in each sample.

### 3.3. Association Analysis

As described above, SNPs were selected before association analyses were conducted. Those SNPs that were homozygous in both YA2016-12 and G184-189 but differed between the two parents, with a read depth >2, were kept; those with an SNP-index <0.3 or >0.7, or a read depth <7 were excluded. Therefore, after screening, 897,550 SNPs were utilized for association analysis by calculating the SNP-index.

SNP-index graphs were generated for the D-pool (Figure 5a) and T-pool (Figure 5b) by plotting the average SNP-index against the position of each sliding window. By combing the information of the SNP-index values of the D-pool and T-pool, a Δ(SNP-index) was calculated and plotted against the genome positions. By examining the Δ(SNP-index) plot, peak regions above the threshold value were defined as regions where the fitted values were greater than the standard deviations above the genome-wide media. Thus, one candidate association region on ChrA03 was then identified (Figure 5c). It was 12.4 Mbp in length, and contained 1225 genes within the region (Table 5). Among these genes, 811 genes contained a non-synonymous SNP, a frameshift InDel, stopgain, or stoploss; therefore, these 811 genes varied between YA2016-12 and G184-189 significantly on the genomic level.

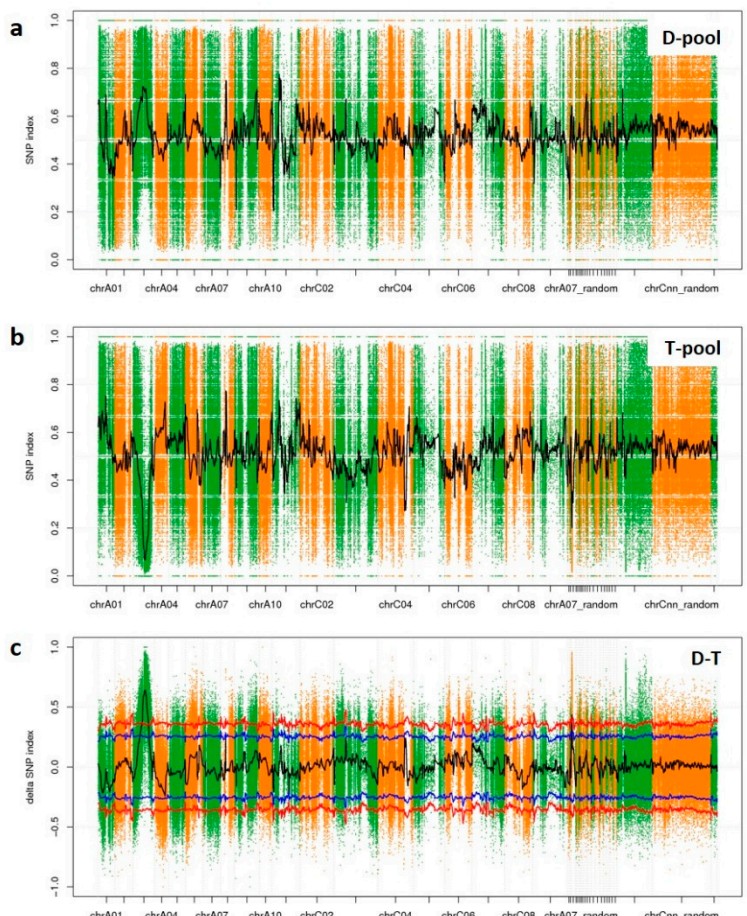

**Figure 5.** SNP-index of D-pool and T-pool and Δ(SNP-index) of D-T on chromosomes. (**a**) SNP-index of D-pool; (**b**) SNP-index of T-pool; (**c**) Δ(SNP-index) of D-T. The peak regions above the threshold values are defined as an associated region. The x axis represents chromosomal position. The y axis shows the SNP-index or Δ(SNP-index). Blue line: the threshold value line for the 95% confidence intervals (*P* value = 5%); red line: the threshold value line for the 99% confidence intervals (*P* value = 1%).

**Table 5.** Basic information about the association region on ChrA03.

| Chromosome | Start Position | End Position | Length (Mb) | Gene Number | +/− DNA Strand |
|---|---|---|---|---|---|
| ChrA03 | 9500001 | 21900001 | 12.4 | 1225 | + |

*3.4. Identification of Candidate Genes by Ortholog Alignment and Gene Annotations*

We first aligned these 811 genes that varied between YA2016-12 and G184-189 with the known 143 plant height-related genes in *Arabidopsis* [30], and selected two genes based on the function of their orthologs in *Arabidopsis* (Table 6): *BnaA03g31770D* (ortholog of AT3G11540) and *BnaA03g37960D* (ortholog of AT5G35750). AT3G11540 [34] and AT5G35750 [35] were reported to regulate plant height in *Arabidopsis*; therefore, their orthologs in rapeseed, *BnaA03g31770D* and *BnaA03g37960D*, are presumed to have similar functions.

In addition, we applied KEGG pathway and GO annotations of the 811 genes and found nine candidate genes, including the two genes mentioned above (*BnaA03g31770D*, *BnaA03g37960D*, *BnaA03g24740D*, *BnaA03g40550D*, *BnaA03g26120D*, *BnaA03g35130D*, *BnaA03g42350D*, *BnaA03g25610D*, and *BnaA03g39850D*) (Table 7). These genes were annotated to the gibberellin-mediated signaling pathway (GO: 0010476) or to the cytokinin metabolic process (GO: 0009690), which are directly involved in plant height, or annotated to other proteins that indirectly interact with gibberellin or cytokinin signaling.

**Table 6.** The candidate genes *BnaA03g37960D* and *BnaA03g31770D* and their orthologs in *Arabidopsis*.

| Gene in *B. napus* | Orthologs in *Arabidopsis* | | |
|---|---|---|---|
| | Gene ID | Gene Name | Gene Function |
| *BnaA03g37960D* | AT5G35750 | *AHK2* | Encodes histidine kinase AHK2 |
| *BnaA03g31770D* | AT3G11540 | *SPY* | Encodes an N-acetyl glucosamine transferase that may glycosylate other molecules involved in gibberellin (GA) signaling. Contains a tetratricopeptide repeat region and a novel carboxy-terminal region. SPY acts as both a repressor of GA responses and as a positive regulation of cytokinin signaling. SPY may be involved in reducing reactive oxygen species accumulation in response to stress. |

The information about orthologs in *Arabidopsis* was according to Shi et al. [30].

**Table 7.** Seven of the candidate genes screened out by annotation information. KOG: Clusters of Orthologous Groups.

| Gene ID | Pathway | Gene Ontology (GO) | KOG-Function Description |
|---|---|---|---|
| *BnaA03g24740D* | | | Suppressor of G2 allele of skp1 |
| *BnaA03g40550D* | | GO: 0009739//response to gibberellin; GO: 0071370//cellular response to gibberellin stimulus; GO: 0010476//gibberellin mediated signaling pathway | Leucine-rich repeat |
| *BnaA03g35130D* | | GO: 0009690//cytokinin metabolic process | tRNA delta(2)-isopentenylpyrophosphate transferase |
| *BnaA03g42350D* | ko04075//Plant hormone signal transduction | GO: 0009735//response to cytokinin; GO: 0009755//hormone-mediated signaling pathway | GATA-4/5/6 transcription factors |
| *BnaA03g25610D* | ko03008//Ribosome biogenesis in eukaryotes | GO: 0031461//cullin-RING ubiquitin ligase complex | WD repeat protein |
| *BnaA03g26120D* | ko04141//Protein processing in endoplasmic reticulum; ko04120//ubiquitin-mediated proteolysis | GO: 0000151//ubiquitin ligase complex; GO: 0043234//protein complex; GO: 0031461//cullin-RING ubiquitin ligase complex; GO: 0044389//ubiquitin-like protein ligase binding; GO: 0031625//ubiquitin protein ligase bindingGO: 0043632//modification-dependent macromolecule catabolic process; GO: 0030163//protein catabolic process; GO: 0019941//modification-dependent protein catabolic process; GO: 0006511//ubiquitin-dependent protein catabolic process | Cullins |
| *BnaA03g39850D* | ko03015//mRNA surveillance pathway | GO: 0000151//ubiquitin ligase complex; GO: 0031461//cullin-RING ubiquitin ligase complex | mRNA cleavage stimulating factor complex; subunit 1 |

## 4. Discussion

Heterosis is extensively utilized in rapeseed breeding in China. Heterosis has resulted in taller rapeseed plants with more biomass, but this increase in plant height has had the undesired effect of enhanced risk of lodging. Lodging can reduce the seed yield [36] and significantly reduce the oil content of the seed, as well as the seed number per pod, seed weight, and other yield-related traits. Moreover, lodging increases the difficulty of mechanized harvesting and decreases the harvesting index [37]. Therefore, dwarf or semi-dwarf rapeseed varieties are considered ideal to increase and stabilize the yield and harvesting index—especially via use of the dwarfism genes produced by mutation, as these genes are known to confer lodging resistance and high yield [9]. The ideal height for mechanized harvesting is semi-dwarf.

The dwarfism trait investigated in this study was originally discovered in *B. napus* × *B. rapa* hybrid offspring. The homozygous line YA2016-12 was obtained after self-pollination for several generations. At the seedling stage, unlike the other plants, the leaves of YA2016-12 were dark green and wrinkled. When the other genotypes (including G184-189) started bolting, YA2016-12 had only procumbent leaves and a shorter stem, showing obviously stunted growth. At the final-flowering stage, YA2016-12 was approximately 50% of the height of G184-189, as well as some other rapeseed materials observed in this study. These phenomena could be due to an endogenous hormone deficiency in the plant.

G184-189 has a tall stature, which made it suitable as the other parent for a cross to study the segregation of the dwarfism trait. All of the $F_1$ individuals showed a semi-dwarf plant stature, which implied that the trait was semi-dominant. Variations in plant height were observed in the $F_2$ population: the number of tall, semi-dwarf, and dwarf individuals satisfied the 1:2:1 segregation ratio, further supporting that the dwarfism trait is controlled by a single semi-dominant allele. Most reported natural dwarfism traits are controlled by multiple minor loci, which reflect continuous variation in a population [30,38–40]. However, other dwarfism traits were found to be controlled by a single allele, especially those discovered from artificial mutation. Foisset et al. [1] used EMS to mutagenize rapeseed and discovered a dwarf line controlled by a recessive allele (bzh/bzh). Zeng et al. [41] identified a recessive gene in a dwarf mutant (*bnaC.dwf*) mutagenized by EMS that was responsible for the dwarfism.

The correlation analysis indicated that plant height was significantly positively correlated with branch segment length (BSL) but not with main inflorescence length or first primary branch height (FPBH), i.e., the dwarfism trait in YA2016-12 is mainly due to the shortened BSL. Here, one advantage of a semi-dominant dwarfism allele was that the hybrid ($F_1$) plant was semi-dwarf, an ideal height for mechanized harvesting. A too low plant height or FPBH would hinder mechanized harvesting; instead, with this dwarfism trait, the BSL is shorter, which results in a more compact plant type. In addition, because this trait is conferred by a single allele, it would be easier to use in breeding.

BSA, also known as linkage disequilibrium (LD) mapping, provided a convenient but effective strategy for identifying molecular markers linked to target genes by genotyping bulked DNA samples from two populations with distinct phenotypes [16]. In this study, instead of traditional markers like SRAP [17], SSRs [18], or RAPD [19], we used SNPs/InDels detected by WGR. SNPs occur at high frequencies in plant genomes, offering larger quantities of data, and are suitable for the construction of high-density genetic linkage maps for crops with large genomes [42,43]. Moreover, compared to reduced-representation sequencing (RRS), WGR produced more data and covered more regions on the reference genome with more sequencing depth. We obtained 225 Gbp of raw data, approximately 339 M HQ clean reads for each sample after filtration, with the average sequencing depth of 37.5×. After screening, 897,550 SNPs qualified for association analysis by the SNP-index method, which provided higher density.

According to the Δ(SNP-index) data, the peak regions above the threshold line revealed a candidate association region on ChrA03 with a length of 12.4 Mbp. Although this region was larger than expected, its segregation fitted the 1:2:1 ratio. Li et al. [4] also identified a locus on ChrA03,

but its position was different from the one identified here. Further analysis revealed that there are 1225 genes within our candidate association region and 811 of them contained sequence variations between YA2016-12 and G184-189.

We aligned these 811 genes to the known 143 plant height-related genes in *Arabidopsis* gathered by Shi et al. [30] and selected two important genes: *BnaA03g31770D* (ortholog of AT3G11540) and *BnaA03g37960D* (ortholog of AT5G35750). AT3G11540 [34], known as *SPINDLY* (*SPY*) in *Arabidopsis*, encodes an O-linked N-acetylglucosamine transferase, which is a negative regulator of gibberellin signaling and activates DELLA proteins [44–47], which are critical in the gibberellin signaling pathway. The *spy-4* mutant in *Arabidopsis* has decreased plant height compared to the wild type [47]. AT5G35750, known as *Arabidopsis Histidine Kinase 2* (*AHK2*), encodes a histidine kinase, which acts as a cytokinin receptor. In *ahk2 ahk3* double mutants of *Arabidopsis*, the shoot growth is reduced [35,48]. *BnaA03g31770D* and *BnaA03g37960D* shared 91% and 76% amino acid identification with AT3G11540 and AT5G35750, respectively; therefore, they are presumed to perform similar functions and are considered important candidate genes.

YA2016-12 resembled some reported dwarf rapeseed plants resulting, to some extent, from GA deficiency, such as wrinkled, dark green leaves and sclerotiniose susceptibility; therefore, using the gene annotations, especially GO annotation and the KEGG pathway, we found seven additional important genes: (1) *BnaA03g24740D* was described as a suppressor of the G2 allele of skp1. Skp1 is a subunit of the Skp1/cullin/F-box (SCF) protein complex, which interacts with and ubiquitylates DELLA proteins so that they can be degraded by the 26S proteasome. DELLA proteins are vital repressor proteins in the gibberellin signaling pathway [8,49,50]. (2) *BnaA03g40550D* was annotated to respond to gibberellin (GO: 0009739), cellular response to gibberellin stimulus (GO: 0071370), and the gibberellin-mediated signaling pathway (GO: 0010476). (3) *BnaA03g26120D* was described as a cullin protein, which is another subunit of the SCF protein complex, combined with its annotations as ubiquitin ligase complex (GO: 0000151) and cullin-RING ubiquitin ligase complex (GO: 0031461). (4) *BnaA03g35130D* is involved in the cytokinin metabolic process (GO: 0009690). (5) *BnaA03g42350D* is related to plant hormone signal transduction (ko04075) and response to cytokinin (GO: 0009735). (6) *BnaA03g25610D* and (7) *BnaA03g39850D* are both related to the cullin-RING ubiquitin ligase complex (GO: 0031461). Although the last four genes need more investigation to support their role in plant height, they are also considered potential candidate genes based on our current information.

## 5. Conclusions

In this study, we investigated a dwarfism trait controlled by a semi-dominant allele in a rapeseed line, YA2016-12. A total of 811 genes with non-synonymous mutation sites or frameshift mutation sites were found in an association region on ChrA03 identified by BSA and WGR. Nine genes involved in gibberellin or cytokinin signal transduction were considered important candidate genes for the dwarfism trait in rapeseed according to the alignment to known plant height-related orthologs in *Arabidopsis*, as well as their KEGG pathway and GO annotations.

Since this trait is controlled by a single semi-dominant allele, the $F_1$ hybrid progeny of the cross between dwarf YA2016-12 and tall G184-189 were of the semi-dwarf phenotype, which is ideal for lodging resistance and mechanized harvesting. The allele identified here would be easier to use in breeding than other dwarfism traits controlled by multiple QTL, demonstrating the potential application of our findings for commercial breeding.

**Supplementary Materials:** The following are available online at http://www.mdpi.com/2073-4395/9/3/129/s1, Table S1: Number of SNPs in various locations for each sample, Table S2: Statistics of different functions of SNPs in each sample, Table S3: Number of InDels in various locations for each sample, Table S4: Statistics of different functions of InDels in each sample.

**Author Contributions:** Conceptualization, L.C. and L.J.; formal analysis, H.L. and L.W.; resources, H.L. and S.Z.; data curation, L.C. and H.L.; writing—original draft preparation, L.C.; writing—review and editing, C.C.; visualization, B.Z. and J.J.; supervision, L.J.; project administration, H.L.; funding acquisition, H.L. and J.Z. All authors reviewed the manuscript.

**Funding:** This research was funded by the Modern Agro-Industry Technology Research System of China, grant number CARS-12; National Key Research and Development Plan, grant number 2016YFD0101305; Major Science and Technology Special Subject of Sichuan Province, grant number 2018NZDZX0003; Scientific Observing and Experimental Station of Oil Crops in the Upper Yangtze River, Ministry of Agriculture, P. R. China, grant number 09203020; Financial Innovation Ability Promotion Project of Sichuan Province, grant number 2016ZYPZ-013; National Key Research and Development Plan, grant number 2018YFD0100500; National Natural Science Foundation of China, grant number 31560402; Sichuan Science and Technology Program, grant number 18ZDYF0623; and Sichuan Crop Breeding Community, grant number 2016NYZ0031.

**Acknowledgments:** We gratefully acknowledge the funding that supports this research.

**Conflicts of Interest:** The authors declare that they have no conflict of interest.

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
