# Peer review of "Rapid Identification of a Genomic Region Conferring Dwarfism in Rapeseed (Brassica napus L.) YA2016-12"

_agronomy, doi:10.3390/agronomy9030129_

Round 1
Reviewer 1 Report
The current manuscript title “Rapid identification of a genomic region conferring a 2 dwarfism in rapeseed (Brassica napus L.) YA2016-12”, the objectives of work dwarf rapeseed line YA2016-12 crossed with the tall rapeseed line G184-189, and F2 population DNA was used to identify the trait-associated regions from the SNP and insertion/deletion (InDel) data. The work is interesting, and it needs substantial improvement needed. I want to suggest some minor points….
1) Minor language improvement needed
2) Figure 3 and Figure 4, need to plot based on difference percent or in another way that reveals the heterogeneity among the different samples.
3) Authors used “Dwarf” and “Tall” in small in all tables and figures, it needs to revise.
Please find the PDF with some minor comments.

Reviewer 2 Report
It would be helpful to mention that rapeseed is an allotetraploid and briefly explain that the A and C chromosome designations refer to the two parent genomes.
How many generations of selfing were carried out in the two parents? How homozygous are they? Could they be segregating for modifier genes?
How phenotypically uniform were the F1 plants? Did the greenhouse environment affect plant height? That is, was plant height the same for F1 plants growing in different locations in the greenhouse?
The frequency distribution in Figure 2 leads the reader to believe that height is a quantitative trait, or at least that it is influenced by environment. However, it seems to be treated as a qualitative trait, based on phenotypic categories. Please expand on this observation.
It doesn't appear as if the trial was repeated, which is necessary to demonstrate that the results are consistent.
The mean of the tall F2 plants is significantly lower than that of the tall parent. This doesn't align with the comment that dwarfism is controlled by a single semi-dominant allele. Please provide an explanation for this observation.
Round 2
Reviewer 2 Report
Comments have been addressed satisfactorily. Thank you.